# A Parallel Feature Fusion Network Combining GRU and CNN for Motor Imagery EEG Decoding

**DOI:** 10.3390/brainsci12091233

**Published:** 2022-09-13

**Authors:** Siheng Gao, Jun Yang, Tao Shen, Wen Jiang

**Affiliations:** Faculty of Information Engineering and Automation, Kunming University of Science and Technology, Kunming 650500, China

**Keywords:** brain-computer interface (BCI), convolutional neural network (CNN), four-class motor imagery, gated recurrent unit (GRU)

## Abstract

In recent years, deep-learning-based motor imagery (MI) electroencephalography (EEG) decoding methods have shown great potential in the field of the brain–computer interface (BCI). The existing literature is relatively mature in decoding methods for two classes of MI tasks. However, with the increase in MI task classes, decoding studies for four classes of MI tasks need to be further explored. In addition, it is difficult to obtain large-scale EEG datasets. When the training data are limited, deep-learning-based decoding models are prone to problems such as overfitting and poor robustness. In this study, we design a data augmentation method for MI-EEG. The original EEG is slid along the time axis and reconstructed to expand the size of the dataset. Second, we combine the gated recurrent unit (GRU) and convolutional neural network (CNN) to construct a parallel-structured feature fusion network to decode four classes of MI tasks. The parallel structure can avoid temporal, frequency and spatial features interfering with each other. Experimenting on the well-known four-class MI dataset BCI Competition IV 2a shows a global average classification accuracy of 80.7% and a kappa value of 0.74. The proposed method improves the robustness of deep learning to decode small-scale EEG datasets and alleviates the overfitting phenomenon caused by insufficient data. The method can be applied to BCI systems with a small amount of daily recorded data.

## 1. Introduction

Brain-computer interface (BCI) technology establishes an information interaction channel between brain and computer, which does not depend on normal peripheral nerves and muscle tissues [1]. It can directly convert the information sent by the brain into commands to drive external devices and replace human limbs or language organs to communicate with the outside world and control external devices. There are three types of BCI according to the position of sensors in the brain. Among them, invasive and partially invasive BCI requires the neurosurgical implantation of collecting electrodes into the cerebral cortex. Although this invasive implantation can obtain a high signal-to-noise ratio (SNR) and high-spatial-resolution signals through the operation of implanted electrodes, it has certain infection risks and safety problems. Non-invasive BCI monitors electrical changes in brain neurons on the scalp via a wearable device attached to the scalp. An EEG cap is the most used non-invasive sensor, which can collect brain signals in the form of electroencephalography (EEG). Since non-invasive BCI requires no surgery, it is safe, non-invasive, easy to operate, and low in cost, and EEG signals have a high temporal resolution. Because of its advantages, non-invasive BCI based on EEG has become a research hotspot [2]. 

EEG-based BCI paradigms mainly include motor imagery (MI), steady-state visual evoked potential (SSVEP), and event-related potential (ERP). The MI paradigm is a mental process that imagines limb movements but does not perform real actions [3]. Compared with SSVEP and ERP paradigms, the MI paradigm does not need external stimulation and belongs to spontaneous EEG. Relevant studies have proven that during the process of motor imagery, the cerebral cortex will produce two types of significantly changed rhythm signals, namely the 8–13 Hz α rhythm and 17–30 Hz β rhythm. When subjects imagine the movement of one limb, the EEG rhythm energy in the contralateral motor perception region of the cerebral cortex decreases significantly, while that in the ipsilateral motor perception region increases. This phenomenon is called event-related desynchronization (ERD) and event-related synchronization (ERS) [4]. Based on this relationship, BCI is widely used in the rehabilitation of neuromotor disorders [5], the treatment of stroke patients [6], and the assistance of limb amputations [7]. BCI contributes to the independent living of disabled patients and improves the lifestyles of healthy users. Currently, there is an increasing number of BCI systems that can operate more complex devices, including auxiliary robots [8], autonomous driving [9], and robotic arms [10].

In general, a typical non-invasive BCI consists of five main processing stages [11]: EEG data acquisition, data preprocessing, feature extraction, pattern recognition, and control of external devices. In these stages, feature extraction and pattern recognition are the keys to ensuring the operation of the whole BCI system. Although EEG-based non-invasive BCI technology has made great progress, its application in practice is constrained by many factors. One factor is the inherent characteristics of the EEG, including low SNR, non-stationary characteristics over time, and artifacts from other brain regions [12]. The other factor is the difficulty in obtaining large-scale EEG data in many practical application scenarios. Small-scale datasets make it difficult to achieve strong robustness and high decoding accuracy in the decoding model. 

To solve these problems, many researchers have studied feature extraction and classification methods in MI-EEG recognition. The traditional power spectral density (PSD) analysis method does not consider the spatial correlation of EEG, and its resolution is limited, so it cannot represent the nonlinear characteristics of EEG signals. Common spatial pattern (CSP) is a spatial feature extraction algorithm for two classes of MI-EEG [13]. It extracts spatial distribution components of each class from multi-channel EEG data. Based on CSP, Kai et al. [14] proposed the filter bank common spatial pattern (FBCSP). They use filter banks to decompose EEG into nine bands in the frequency range of 4–40 Hz at a bandwidth of 4 Hz. Then, the CSP algorithm is applied to each sub-band to obtain highly separable features. In addition, improved algorithms based on CSP have been proposed, such as the common spatio-spectral pattern (CSSP) [15], common sparse spectral spatial pattern (CSSSP) [16], and sub-band common spatial pattern (SBCSP) [17]. A large number of studies have proven that this series of CSP-based spatial feature extraction algorithms can achieve good performance in many MI recognition tasks. Gaur et al. [18] used sliding windows to capture multi-segment EEG signals and used CSP and linear discriminant analysis (LDA) to extract and classify the features of the signals in each time window. Ko et al. [19] extracted the features of MI-EEG by fast Fourier transform (FFT) and CSP and then input the features into a multi-modal fuzzy fusion framework to decode EEG signals. Zhang et al. [20] designed a temporally constrained sparse group spatial pattern (TSGSP), which can simultaneously optimize the filtering band and time window in CSP and improve the classification accuracy of MI-EEG. Jin et al. [21] extracted multi-channel EEG features using regularized CSP (RCSP) and then decoded MI-EEG using a support vector machine (SVM) with a radial basis function (RBF) kernel. Ang et al. [22] verified the performance of FBCSP on the BCI Competition IV Datasets 2a and 2b, and they proposed one-versus-rest (OVR), divide-and-conquer (DC), and pair-wise (PW) approaches for decoding four classes of MI-EEG data.

In recent years, deep learning (DL) has made great progress in the fields of computer vision, speech recognition, natural language processing, and data generation. When decoding EEG data using machine learning (ML) methods, some prior knowledge and experience is required to extract features, which leads to limited accuracy in EEG decoding and is costly and time-consuming [23]. The neural network is the main architecture of deep learning. Because of its end-to-end structure, it can greatly reduce the need to manually extract EEG features. Meanwhile, it can overcome the limitation of prior knowledge and effectively learn potential features from EEG data. By using neural networks, non-neuromedical researchers can conduct EEG data analysis. Tabar et al. [24] sequentially combined convolution neural network (CNN) and stacked autoencoder (SAE) models to decode MI-EEG. The network can automatically extract the time, frequency, and channel information from EEG time-frequency images. Dai et al. [25] explored the influence of convolution kernels of different sizes on MI-EEG decoding performance. Lawhern et al. [26] designed the EEGNet model, which encapsulates the concept of feature extraction in EEG decoding. Liu et al. [27] reconstructed the raw EEG data into a three-dimensional representation to better express the spatio-temporal features of EEG. Meanwhile, they constructed a densely connected multi-branch 3D CNN to decode 3D representations of MI-EEG data. Yang et al. [28] proposed a two-branch time-frequency convolution neural network (TBTF-CNN) to extract multiple features of EEG synchronously. Hou et al. [29] combined the EEG source imaging (ESI) method with joint time-frequency analysis to classify MI tasks.

At present, great progress has been made in decoding two classes of MI tasks based on deep learning methods. However, with the increasing number of MI tasks and EEG channels, decoding research for four classes of MI tasks (left hand, right hand, tongue, and foot) needs to be further explored. Most studies only extract single spatio-temporal features, and few apply feature fusion methods from other fields to the four classes of MI-EEG. In addition, most studies tend to adopt a serial structure when decoding four-class MI tasks, stacking a CNN and recurrent neural network (RNN), to extract temporal, frequency, and spatial features [30]. This ignores a large amount of valid information in the middle layers of the neural network, resulting in poorer classification performance. Research has shown that information hidden in the middle layer of the neural network can help to improve the discrimination of the model [31]. More importantly, it is very difficult to collect large-scale EEG data, and all publicly available MI-EEG datasets are small-scale datasets. When the training data are limited, the neural networks tend to suffer from problems such as overfitting and poor robustness. In this study, we propose a data augmentation method for MI-EEG. As EEG is a temporal signal, it has high temporal resolution. Thus, we slide the raw EEG horizontally along the time axis and recombine it to generate more data. The new data preserve the temporal and spatial distribution of the original data and effectively expand the original dataset. In addition, we design a parallel-structured feature fusion model using neural network tools. The model is composed of a GRU and CNN in parallel, which avoids the mutual interference of spatio-temporal features and does not lose the feature information of the middle layer. The advantages of using GRU and CNN in parallel are as follows. First, the original EEG has a clear representation of temporal features and a poor representation of spatial and frequency features. Due to its directional cycle mechanism, the GRU network can effectively extract the temporal features of time series [32]. Therefore, for the GRU part of the model, we input the time-series representation of EEG signals. Secondly, we input the spectrum representation of EEG into the CNN part. An EEG signal is similar to a speech signal, which contains key information in the frequency domain. A spectrogram can better represent the spatial and frequency features of EEG. The CNN network has powerful spatial feature extraction ability for two-dimensional data, especially in image form [33]. Therefore, the network can effectively extract the detailed spectral features of space and frequency by inputting the spectrogram into the CNN. In Section 3.2 of this paper, we describe ablation experiments to further explore the influence of different input forms on decoding performance.

The main contributions of this paper are summarized as follows. (a) A data augmentation method for MI-EEG is designed based on the temporal features of the original EEG. The method preserves the temporal and spatial features of the original EEG. This effectively expands the original dataset and improves the robustness of the decoding model. (b) The influence of parallel-structured neural networks on decoding performance is explored for multichannel, four-class motion imagery data. We combine GRU and CNN to design a parallel-structured feature fusion network (GCFN) and set up ablation experiments to evaluate the influence of different inputs on the decoding performance. (c) Various performance metrics and feature distributions are visualized for four classes of MI datasets. The effectiveness of the proposed data augmentation method is verified by setting up machine learning comparison experiments.

## 2. Methods

### 2.1. System Architecture

Figure 1 shows the system architecture of the proposed method. When the subjects perform the MI task, the corresponding potential changes will occur in the motor perception area of the brain. This is manifested in the increase and decrease in energy in the α band (8–13 Hz) and the β band (17–30 Hz). The BCI system records the EEG signals of the subjects through wearable devices and saves the data in the two-dimensional form of electrode × sampling points. In this paper, data augmentation is first performed on raw EEG data to ensure the robustness of the deep learning model. In the data preprocessing stage, a fifth-order Butterworth bandpass filter is designed to filter MI-EEG data, and the target frequency band of 8–30 Hz is obtained. Subsequently, a continuous wavelet transform (CWT) is applied to the MI-EEG data from 8 to 30 Hz to obtain the time-frequency image of each channel. Then, the filtered EEG data are normalized using the z-score method. The spectral data are then divided by 255 for each pixel value. This eliminates the negative impact of outliers on the model training and speeds up the convergence of the neural network. In the decoding model, the time-series data and time-frequency image data of MI-EEG are simultaneously input into the two branches of the model. The CWT-CNN branch extracts frequency and spatial features from the 2D time-frequency image, while the EEG-GRU branch extracts temporal features from MI-EEG. Afterward, the extracted features are concatenated into a 1D vector by the GCFN model. Finally, the fusion features are input into a classifier composed of a full connection layer, and the prediction labels are output, thus completing the decoding of MI-EEG.

### 2.2. Data Augmentation Method

In recent years, deep learning has made great progress in image, speech, and natural language processing, mainly attributed to the vast datasets available in these fields. Massive training data can ensure the robustness and classification accuracy of the neural network. However, in many practical application scenarios, it is very difficult to obtain large-scale EEG data. Small-scale datasets can easily lead to the overfitting of neural networks, which affects the robustness and decoding accuracy of the model.

In this study, a data augmentation method is proposed for MI-EEG to alleviate the overfitting problem of the model. Data augmentation has been proven to be effective in many fields, such as computer vision. More training data can be generated by rotation, translation, and cropping to alleviate the overfitting problem and improve the accuracy and robustness of the model [34]. It can be seen from Figure 2 that raw MI-EEG data are usually composed of a 2D matrix in the form of C × T. The rows of the matrix store the data in the channels, and the columns store the data recorded at each sampling point. This 2D representation gives the EEG data a high temporal resolution while preserving the spatial features of the electrode positions. The proposed data augmentation method consists of two phases. (1) Sliding along the time axis. The original EEG data of C × T slide horizontally along the time axis at a step of S. At this point, the data are divided into two segments, namely 0—(T-S) and (T-S)—T. (2) Time series recombination. Exchange the sequence of two pieces of data and slide (T-S)—T pieces of data to the starting point. Phases (1) and (2) are repeated k times until kS ≥ T. At this time, the newly generated data overlap with the original data and stop sliding. Compared with the sliding window method, the new data generated by this method preserve the temporal and spatial information of the original EEG. Meanwhile, by setting an appropriate step size S, new data with certain differences from the original data can be generated, which helps the model to learn more temporal and spatial features and reduce overfitting effectively.

### 2.3. MI-EEG Feature Representation

Reasonable EEG feature representation is conducive to improving the accuracy and efficiency of the whole decoding process. Some researchers use topological maps to represent MI-EEG to improve the spatial resolution between different channels [35]. However, this method will increase the computational complexity of model decoding. According to a literature survey [36], around 30% of studies use EEG in a 2D matrix format as input, and around 30% of studies use spectral images as input. This paper uses the filtered raw signal values and time-frequency images as the input of the deep learning model, and this is the most used EEG representation. 

Figure 3 shows the time-frequency feature expression method of MI-EEG. The time-frequency image of raw EEG can be obtained according to Equations (1)–(3):(1)$$\begin{array}{c} \psi \left(t\right)=e^{i\omega t}\times e^{-\frac{t^{2}}{2}}, \end{array}$$
(2)$$\begin{array}{c} \psi_{\alpha ,\tau }\left(t\right)=\frac{1}{\sqrt{\alpha }}e^{i\omega \frac{\left(t-\tau \right)}{\alpha }}\times e^{-\frac{{\left(t-\tau \right)}^{2}}{2\alpha^{2}}}, \end{array}$$
(3)$$\begin{array}{c} \bar{\Psi }\left(\alpha ,\tau \right)=\frac{1}{\sqrt{\alpha }}\int_{-\infty }^{+\infty }f\left(t\right)\psi_{\alpha ,\tau }\left(t\right)dt. \end{array}$$
where ω is the wavelet center frequency, α is the scale coefficient, and τ is the translation coefficient. This paper uses the “morlet” wavelet as the mother wavelet, and each channel only retains the signal in the frequency band of 8–30 Hz. Then, the time-frequency diagram of each channel is combined into a 2D image according to the electrode order.

### 2.4. Proposed GCFN Architecture

CNN is a feedforward neural network with convolution calculation and depth structure. It can learn representations and is very suitable for processing image data [33]. The convolution operation performs inner products on the input data and the convolution kernel, and the feature map output can be expressed as:(4)$$h_{ij}^{k}=f\left(a\right)=f\left({\left(W^{k}\times x\right)}_{ij}+b_{k}\right)$$
where x is the input data, $W^{k}$ is the weight matrix, $b_{k}$ is the bias vector, ∗ represents the convolution operation, and f(·) represents the activation function.

RNN has memory ability and has certain advantages in learning the nonlinear features of sequence data. The gated recurrent unit (GRU) is a variant of RNN that can effectively alleviate the gradient disappearance and gradient explosion problem in the traditional RNN during training. GRU simplifies the long short-term memory (LSTM) network structure and has fewer parameters [32]. The GRU model considers both historical information $h_{t-1}$ and new information $x_{t}$ when calculating the current state value $h_{t}$, as shown in Equations (5)–(8):(5)$$r_{t}=\sigma \left(W_{r}\cdot \left[h_{t-1},x_{t}\right]\right)$$
(6)$$z_{t}=\sigma \left(W_{z}\cdot \left[h_{t-1},x_{t}\right]\right)$$
(7)$${\tilde{h}}_{t}=tanh\left(W_{\tilde{h}}\cdot \left[r_{t}\times h_{t-1},x_{t}\right]\right)$$
(8)$$h_{t}=\left(1-z_{t}\right)\times h_{t-1}+z_{t}\times {\tilde{h}}_{t}$$
where $r_{t}$ denotes a reset gate; $z_{t}$ denotes an update gate; $W_{r}$, $W_{z}$, and $W_{\tilde{h}}$ denote the weight parameters of the GRU network; σ(·) denotes a sigmoid function.

Figure 4 shows the architecture of the proposed GRU-CNN feature fusion network. In the input part of CWT-CNN, a 2D time-frequency image is reconstructed into a 3D tensor form of 224 × 93 × 1, representing a gray-scale graph with a size of 224 × 93. The features of time-frequency image data are learned by a CNN, which consists of a convolution layer, an activation function layer, a max-pooling layer, and a flattened layer. Inspired by Tarbar et al. [24], this paper uses 1D filters to extract features. Since the temporal, frequency, and electrode position information are used together in the time-frequency image, the 1D convolution kernel sliding along the time axis can extract the three features better. The size of the 1D filter is 224 × 1, and the convolution step is 1. After the convolution calculation, an activation function, “*ReLU*”, is used to output the convolution result, and its expression is as follows:(9)$$f\left(a\right)=ReLU\left(a\right)=\{\begin{array}{l} a,\;\\ 0, \end{array}\begin{array}{c} \;if\;a\geq 0\\ \;if\;a<0 \end{array}$$
where a is defined in Formula (4). Then, the max-pooling layer performs down-sampling processing on the output feature map to extract the identification features. The pooling size and step are set to 1 × 3. The flatten layer converts the output feature map into a 1D vector.

In the input part of EEG-GRU, this paper reorganizes the EEG data into the expression of T × C for input into the GRU network, where T corresponds to the time step of GRU, and C represents the feature number of each time step, i.e., the channel for temporal EEG data. The GRU branch is used to extract the temporal features as the Appendix A. The branch consists of two stacked GRU layers, with 25 units in the first layer and 50 units in the second layer. Through Equations (5)–(8), the GRU extracts the temporal features of EEG data and outputs a 1D vector.

The feature fusion layer fuses the feature vectors into a 1D vector and then inputs the vector to the classifier. The classifier consists of several stacked, fully connected layers (FC). The neurons in different layers are fully connected, and the neurons in the same layer are independent. The data passing through the FC layer can be represented as:(10)$$y=f\left(W_{y}x+b_{y}\right)$$
where $W_{y}$ is the weight matrix, $b_{y}$ is the bias vector, and f(·) is the activation function. The first fully connected layer has a total of 128 neurons and uses the “*ReLU*” activation function. The second fully connected layer has 4 neurons as outputs, and the output y of the “softmax” activation function is mapped to the prediction probability $y_{p,m}$, which is expressed as follows:(11)$$y_{p,m}=\frac{e^{y_{m}}}{\sum_{m}^{T}e^{y_{m}}}$$
where m is the index of y, and T represents the total number of classes. Moreover, the cross-entropy loss function is used to measure the prediction results and true values, and it is calculated as follows:(12)$$L\left(y_{p},y_{l}\right)=-\underset{m}{\sum }y_{p,m}lo\log y_{l,m}$$
where $y_{p}$ represents the predicted label and $y_{l}$ represents the true label.

In the input stage, the time-frequency image and EEG data are normalized to enhance the data concentration and accelerate the convergence of the neural network. In this step, the z-score standardization method is used for the time-series EEG data and is calculated as follows:(13)$$z_{i}=\frac{x_{i}-\mu }{\sigma }$$
where μ is the mean of the samples for a trial and σ is the standard deviation of the samples. For the spectrogram data, each pixel value was divided by 255 and the formula was calculated as follows:(14)$$x_{ni}=\frac{x_{pi}}{255}$$
where $x_{pi}$ denotes the spectrogram of one trial with a maximum pixel value of 255. $x_{ni}$ denotes the normalized spectrogram data. Meanwhile, the “Dropout” probability of the fully connected layer is set to 0.3 to alleviate the overfitting problem of the model. Moreover, this paper uses the Adam optimizer to update the parameters, thus minimizing the cross-entropy loss function. The detailed parameters of the network are presented in Table 1.

## 3. Dataset and Results

### 3.1. Experimental Dataset

The BCI Competition IV 2a public dataset provided by the Graz University of Technology was used in this paper to evaluate the performance of the proposed method [37]. This dataset contains EEG data from nine healthy subjects. The EEG data were recorded with 22 Ag/AgCl electrodes under the sampling frequency of 250 Hz, and bandpass filtering was performed from 0.5 Hz to 100 Hz. Figure 5 shows the electrode montage corresponding to the international 10–20 system. The whole EEG data collection process is easy and comfortable, and there are no ethical issues. Each subject performed the MI task on two different days, so the EEG data were recorded in two different sessions. Each session involves routine MI tasks that are composed of four classes: left hand, right hand, foot, and tongue. There are 288 motor imagery trials in each session, with 72 trials in each class. Figure 6 shows the timing scheme for each trial. At the beginning of the experiment, a fixed cross “+” and a short sound stimulus were used as cues. When t = 2 s, a prompt appeared on the screen in the form of an arrow pointing to the left, right, below, or above, and it lasted for 1.25 s on the screen. Then, the subjects were asked to perform the corresponding motor imagery task within 4 s. When t = 6 s, the experiment of motor imagery ended, and the subjects rested for 1.5 s. After the data were collected, experts evaluated the influence of artifacts on the data and marked the trial containing artifacts as a “1023” event. The details of the dataset are shown in Table 2.

### 3.2. Performance of the Proposed GCFN

In this paper, 10-fold cross-validation is adopted to validate the decoding performance of the proposed model. The dataset is divided into 10 equal subsets, and 90% of them are randomly selected as training data and the remaining 10% as validation data. According to the MI-EEG acquisition paradigm introduced in Section 3.1, this paper selects 0.5–4 s EEG data after the MI task starts. Meanwhile, the raw data are expanded by the data augmentation method described in Section 2.2, and the sliding step S is set to 80. Thus, the dataset of each subject can be represented as 6336 × 22 × 875, where 6336 denotes the number of trials, 22 denotes the number of electrodes, and 875 corresponds to the sampling point of 3.5 s. The whole experimental platform is implemented with the famous DL architecture of TensorFlow 2.0, and the NVIDIA RTX 3080 GPU (NVIDIA, Santa Clara, CA, USA) is employed to accelerate the model training process.

In this paper, ablation experiments are described to evaluate the effect of different branches on decoding performance. The model structure under three different conditions is shown in Figure 7. The implementation details include the following: (a) EEG-GRU—only EEG data are input, and the time series features of the EEG data are extracted by two stacked GRU units and then input into the classifier; (b) CWT-CNN—only time-frequency images are input, and the time, frequency, and spatial features of the EEG data are extracted by a 1D convolution kernel with a size of 224 × 1; the features are processed by the max-pooling layer and flatten layer and then input to the classifier; (c) full model—the GRU-CNN feature fusion network is constructed following the method described in Section 2.4. Specifically, the EEG and time-frequency images are used as inputs to fuse the features of different branches, and then the fused 1D feature vectors are input to the classifier.

Figure 8 illustrates the ablation experiment results of nine subjects. It can be seen that the global accuracy of the EEG-GRU model is 62.9%. Except for subjects 6 and 9, the average classification accuracy of EEG-GRU for a single subject is lower than that of CWT-CNN. The results show that the EEG-GRU model obtains poor performance due to only learning temporal features. The average classification accuracy of the CWT-CNN model is 72.7%. Compared with EEG-GRU, CWT-CNN has improved the classification accuracy in most subjects. The experimental results show that the CNN model is effective in decoding EEG. As a representation of EEG, a time-frequency image contains time, frequency, and spatial information, which is conducive to improving the decoding performance. The GCFN model achieves an average classification accuracy of 80.7%, which is the best among the three models, and the test results on a single subject are greatly improved. The experimental results verify the effectiveness of the proposed model. According to the results of ablation experiments, the CNN can extract the features of time-frequency images better, which helps to improve EEG decoding. This paper visualizes the time-frequency image and feature maps of subjects 3 and 8, as shown in Figure 9. It can be seen that the time-frequency image reflects the relationship between the channel, time, and frequency of the EEG signal, so most studies using spectrograms to decode EEG have achieved good performance. By comparing the time-frequency image before and after convolution, only some areas of the feature map are highlighted. In addition, this paper maps the MI-EEG features of nine subjects after the GCFN model to a 2D plane to observe the feature distribution. As shown in Figure 10, the red, blue, green, and orange colors represent the MI-EEG features of the left hand, right hand, foot, and tongue, respectively. It can be seen from the scatter diagram of feature distribution that the four classes of MI feature distribution of subjects 1, 3, 7, 8, and 9 have certain discrimination. Meanwhile, the four classes of MI feature of subjects 2, 4, 5, and 6 are not distinguished significantly, and their average classification accuracy is lower than 80%. This may be due to the distraction caused by external interferences when performing MI tasks, which caused these subjects to fail to perform the MI tasks effectively.

As shown in Figure 11, the confusion matrices of different models are used for evaluation. In the figure, the diagonal lines represent the global classification accuracy of various tasks, and precision and recall rate are included. As shown in the figure, the recall rate of the left hand, right hand, foot, and tongue in the GCFN model is 82.2%, 77.6%, 83.0%, and 80.0%, respectively. The recall rate of the four classes of MI tasks in the proposed model is higher than those of the other two models. In addition, the precision of the GCFN model is 79.3% for the left hand, 80.2% for the right hand, 80.9% for the foot, and 82.8% for the tongue. In terms of precision, GCFN also performs better than the other two models.

In addition to the average classification accuracy and confusion matrix, the Kappa coefficients of nine subjects under different models are calculated, as shown in Figure 12. The Kappa coefficient is an index to measure consistency in statistics [38], and it can be calculated by the following formula:(15)$$k=\frac{p_{0}-p_{e}}{1-p_{e}}$$
where $p_{0}$ represents the classification accuracy and $p_{e}$ is the chance coincidence rate. Because the dataset contains four classes of MI tasks, the $p_{e}$ value is set to 0.25. As shown in Figure 12, the EEG-GUR model obtains the lowest Kappa value on most subjects, with an average Kappa value of 0.51. The Kappa value of CWT-CNN is 0.13 higher than that of EEG-GRU. The Kappa value of the proposed GCFN model is 0.74, which is significantly higher than that of other models. This indicates that the decoding results of the GCFN model are highly consistent with the actual results.

In addition, a machine learning contrast experiment is conducted to evaluate the performance of the proposed model. CSP is a commonly used feature extraction algorithm in two classes of MI tasks, and it can extract the spatial distribution components of each class from multi-channel EEG data. According to the “OVR” strategy proposed by Ang et al. [22], this paper fine-tunes the CSP algorithm to enable spatial feature extraction for four classes of MI-EEG data. CSP-LDA is the most used MI-EEG decoding model and has achieved good performance in previous experiments. In this paper, the decoding performance of CSP-LDA was evaluated using the EEG data after data augmentation with 10-fold cross-validation. The experimental results are shown in Figure 8, Figure 11, and Figure 12. The average accuracy of the CSP-LDA model is 73.1%, and the Kappa value is 0.64. From the confusion matrix, the precision of the four classes of MI tasks is 71.8%, 72.7%, 73.1%, and 74.6%, respectively. Compared with the GRU-EEG model, the average classification accuracy of the CSP-LDA model is improved by 10.2%. The proposed GCFN model achieves the best accuracy, which is 7.6% higher than that of the CSP-LDA model. Figure 13 shows the MI-EEG features after CSP. It can be seen that the manual features still have some limitations, especially for different individuals.

### 3.3. Comparison with Other Published Results

In this section, the performance of the proposed method is compared with several other methods. Table 3 presents the classification accuracy of nine subjects in the Competition IV 2a dataset under different methods. Meanwhile, the experimental results of the CSP-LDA model are summarized in the table as a baseline. The “*” symbol in the table indicates the classification accuracy calculated according to the Kappa value. It can be seen from the table that the proposed method achieves the highest average classification accuracy and the best performance on most subjects. Specifically, a significant improvement in accuracy is obtained for subjects 4 and 6, which is 7.5% and 13.6% higher than the sub-optimal results, respectively.

## 4. Discussion

It can be seen from the average classification accuracy (Figure 8) that GRU obtains the worst performance among all models. Although EEG data have a high temporal resolution, they also contain a lot of potential information about MI, such as electrode distribution and frequency information. Because of the particularity of its structure, the RNN has great advantages in processing sequence data. However, the RNN only extracts the temporal information of EEG and ignores the spatial and frequency information related to electrodes. By contrast, some studies use CNN to learn the features of the original EEG. Moreover, the spatio-temporal features of the EEG data can be effectively learned by using a 1D filter with different sizes. The average classification accuracy of the CNN model and baseline CSP-LDA model is not significantly different. However, for some subjects (2, 5, 7), the decoding performance using the CSP algorithm is better than that when using the CWT-CNN model. This shows that the machine learning method of manually extracting features is effective for different subjects. Secondly, as a representative form of EEG, the time-frequency image can clearly express various features of EEG. Therefore, most MI-EEG decoding methods based on time-frequency images and CNN perform well. In addition, the end-to-end deep learning model simplifies the decoding of MI-EEG, which provides a basis for online BCI implementation. The proposed GCFN model achieves the highest average classification accuracy in all comparative experiments, which proves the effectiveness of the parallel model and feature fusion. The feature distribution scatter diagram (Figure 10 and Figure 13) indicates that the subjects with a higher classification accuracy have significant discrimination in the four classes of MI features. For subject 1, regardless of whether the CSP algorithm or neural network is used, a few features of the foot and tongue still cannot be well distinguished. Similarly, this phenomenon can be observed for subject 7 on the left hand and right hand. For subjects 4, 5, and 6, with a poor feature distribution, the proposed GCFN model can distinguish the four classes of MI features to a certain extent, while CSP cannot distinguish them well. The results show that, attributed to the ability to automatically learn the potential features of data, the deep learning model is very suitable for processing complex EEG signals. According to the statistical results of the confusion matrix (Figure 11), the recall rate and precision of the four classes of MI tasks in different models are relatively average. All the indexes of the proposed GCFN model are better than those of other models, and the precision distribution is uniform. Moreover, the GCFN model achieves a higher Kappa value (i.e., 0.74) than other models (Figure 12). The experimental results further prove the robustness of the GCFN model. Compared with other published results (Table 3), Ang et al. [22] and Xie et al. [39] designed a complex manual feature calculation method to decode four classes of MI-EEG, and the results were 67.8% and 75.5%, respectively. Mahamune et al. [40] and Sakhavi et al. [41] developed a deep learning decoding model, which achieved a classification accuracy of 71.2% and 74.5%, respectively. Qiao et al. [30] designed a serial neural network model that stacked CNN and GRU to extract spatio-temporal features of EEG. On the same dataset, they achieved an average classification accuracy of 76.6%. However, the serial structure ignores the effective information in the middle layer and may easily cause spatio-temporal features to interfere with each other. Although the MI-EEG decoding performance of the machine learning method and deep learning method is not significantly different, the machine learning method requires complex feature extraction calculations, and the quality of manual features will affect the decoding performance. Deep learning improves the decoding efficiency of MI-EEG with its end-to-end structure. The results of the baseline model and GCFN model show that the proposed data augmentation method can improve the decoding performance of MI-EEG.

## 5. Conclusions

The lack of large-scale EEG datasets limits the application of deep learning in medical rehabilitation. This paper proposes a data augmentation method for EEG to alleviate the overfitting problem of the deep learning model during training. Meanwhile, this paper explores the decoding performance of a parallel deep learning model for MI-EEG and compares the machine learning method of manual features with the end-to-end deep learning method. The experimental results indicate that the proposed GCFN model achieves a better average accuracy of 80.7%. The proposed method not only expands the data scale of EEG but also provides a new approach for improving BCI performance. In future work, with the improvement of hardware, we will apply the proposed GCFN model to online BCI systems to verify its effectiveness and robustness.

## 6. Future Work

In future work, with the improvement of hardware, we will apply the proposed GCFN model to online BCI systems to verify its effectiveness and robustness. Inspired by the literature [42], in subsequent work, we will combine signal decomposition methods and deep learning models to further explore the decoding of four-class MI tasks. In addition, we will design an experimental paradigm to acquire MI datasets from multiple volunteers using EEG caps to ensure the generalization performance of BCI decoding.

## Figures and Tables

**Figure 1 brainsci-12-01233-f001:**
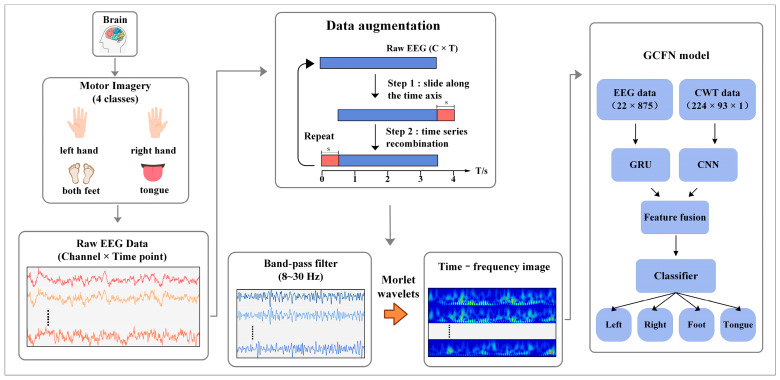
The system architecture of the proposed method, including EEG acquisition, data augmentation, preprocessing and feature extraction, and a deep learning model.

**Figure 2 brainsci-12-01233-f002:**
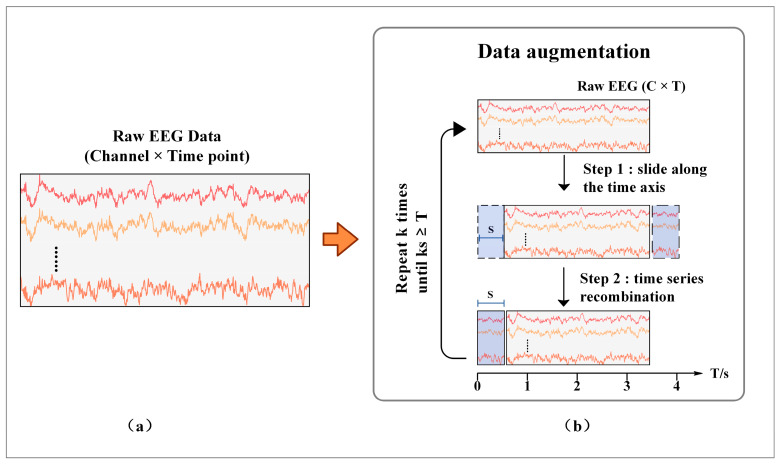
The proposed data augmentation method. (**a**) Representation form of the raw EEG; (**b**) the process of data augmentation.

**Figure 3 brainsci-12-01233-f003:**
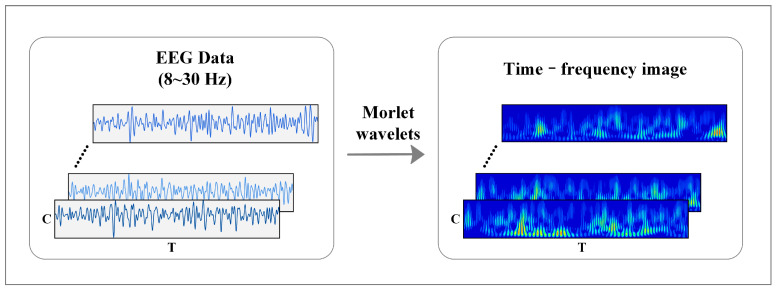
Feature representation of time-frequency image in MI-EEG.

**Figure 4 brainsci-12-01233-f004:**
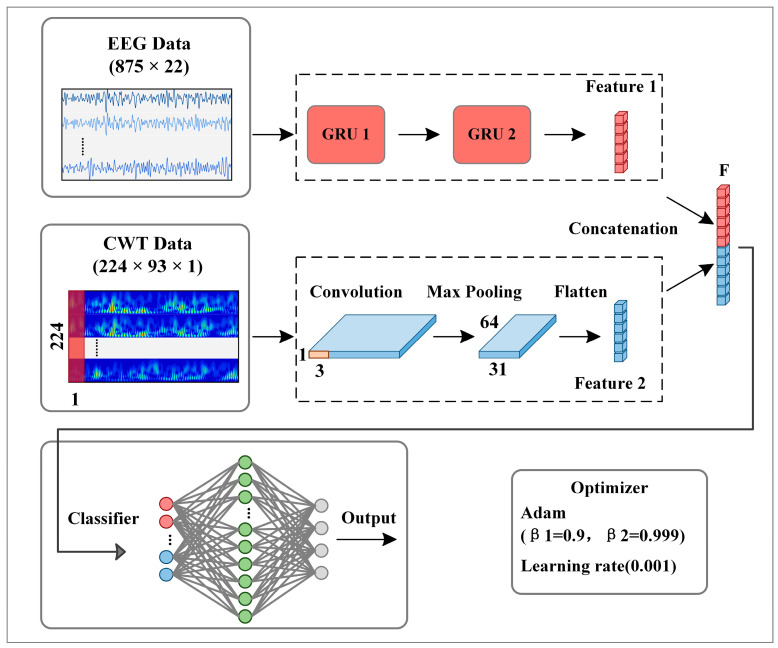
The proposed parallel GRU-CNN feature fusion network architecture.

**Figure 5 brainsci-12-01233-f005:**
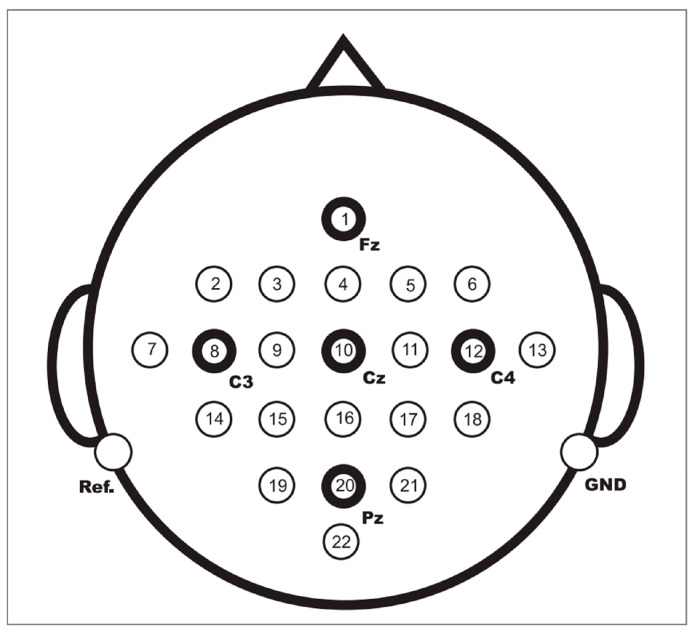
Electrode montage corresponding to the international 10–20 system.

**Figure 6 brainsci-12-01233-f006:**
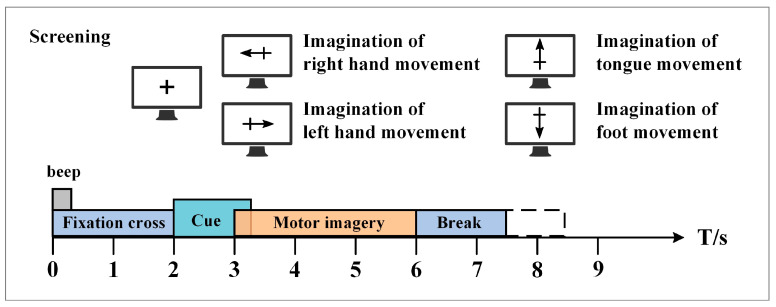
The timing scheme schedule of each MI trial in the BCI competition IV 2a dataset.

**Figure 7 brainsci-12-01233-f007:**
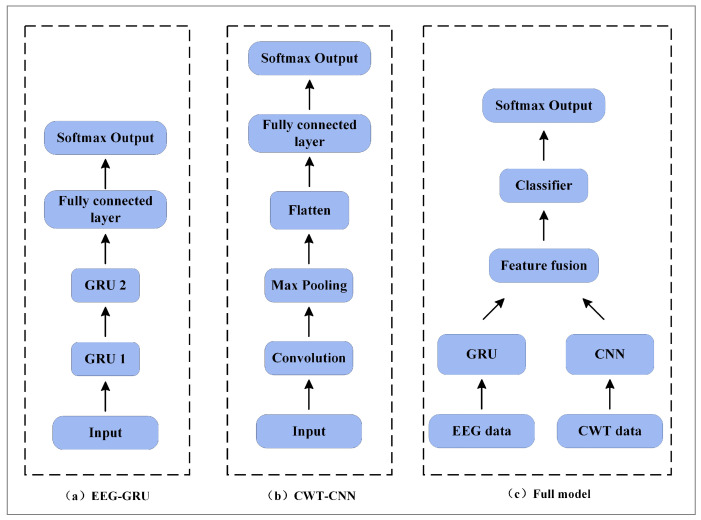
Network model of the ablation experiments.

**Figure 8 brainsci-12-01233-f008:**
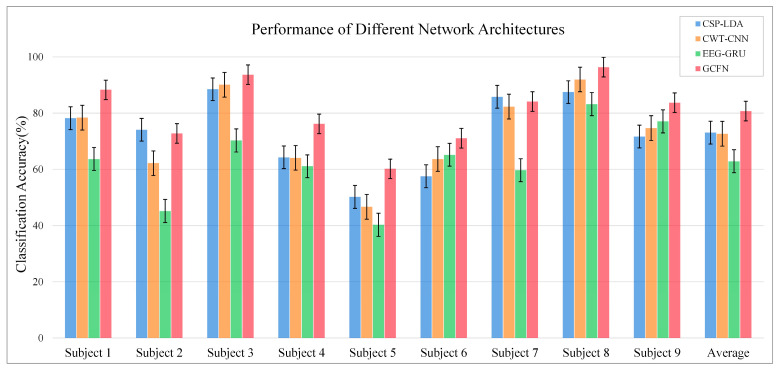
Average classification accuracy of each subject.

**Figure 9 brainsci-12-01233-f009:**
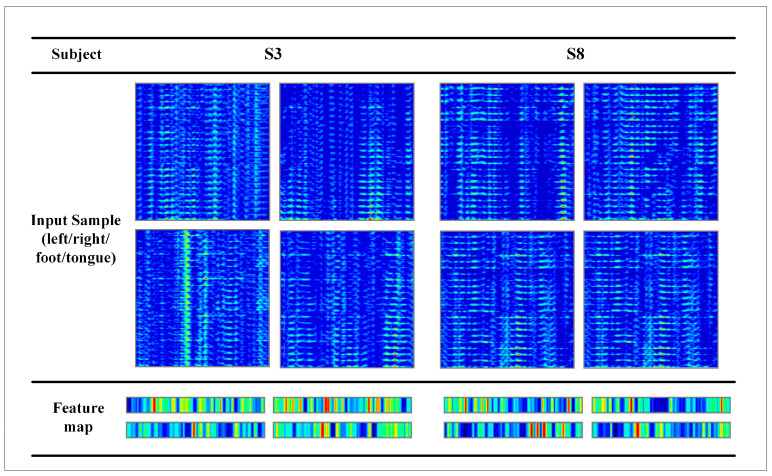
Visualization of the feature maps of randomly selected subjects.

**Figure 10 brainsci-12-01233-f010:**
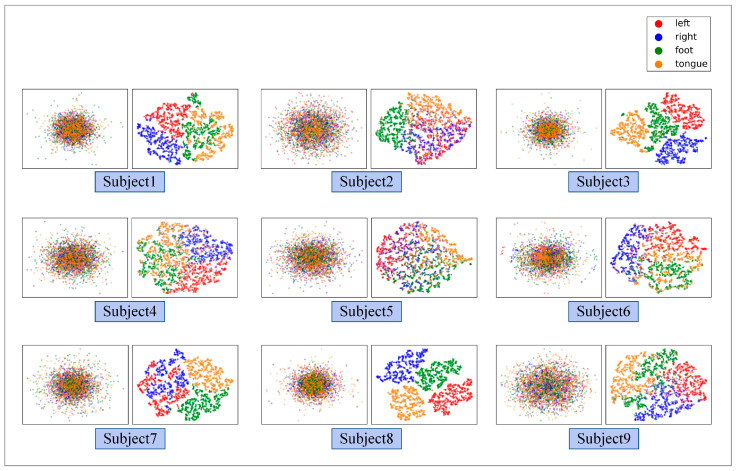
The scatter plot of the feature distribution of 9 subjects under the GCFN model.

**Figure 11 brainsci-12-01233-f011:**
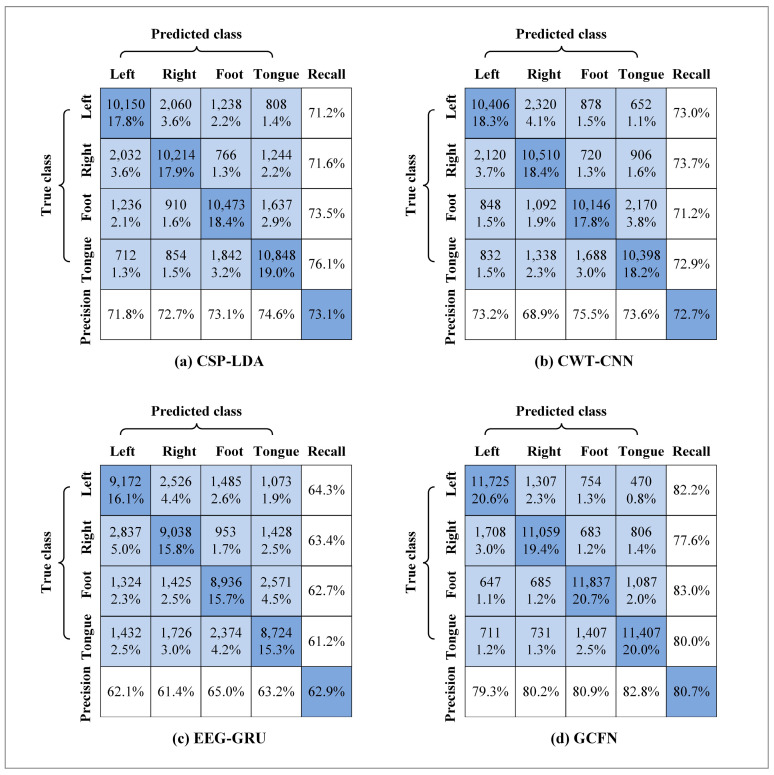
Confusion matrix for different models.

**Figure 12 brainsci-12-01233-f012:**
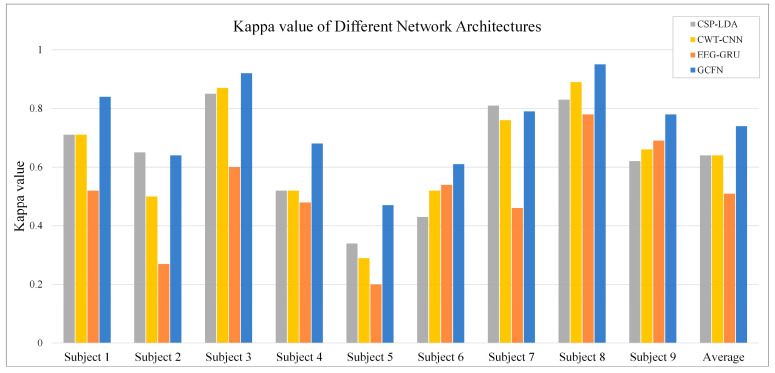
The average Kappa value of each subject under different models.

**Figure 13 brainsci-12-01233-f013:**
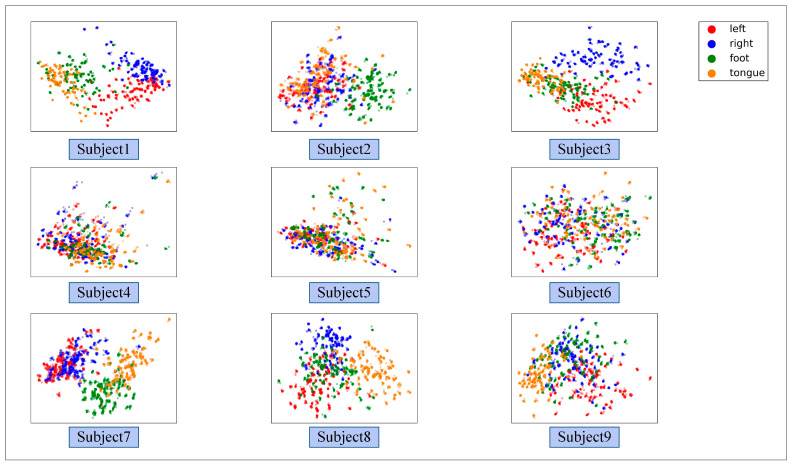
The scatter plot of feature distribution of 9 subjects under the CSP-LDA model.

**Table 1 brainsci-12-01233-t001:** Implementation details for proposed GCFN architecture.

	Layer Type (EEG/Image)	Units	Kernel Size	Stride	Output	Parameters
**CNN**	Input				224 × 93 × 1	
	Conv2D	64	224 × 1	1 × 1	1 × 93 × 64	14,400
	ReLU					
	Max-pooling		1 × 3	1 × 3	1 × 31 × 64	
	Flatten layer				1984	
**GRU**	Input				875 × 22	
	GRU1	25			875 × 25	3675
	Tanh					
	GRU2	50			50	11,550
	Tanh					
**Fusion**	Concatenation				2034	
**Classifier**	FC layer	128				260,480
	ReLU					
	Dropout layer				*p* = 0.3	
	FC layer	4				516
	Softmax					

**Table 2 brainsci-12-01233-t002:** The details of the experimental dataset.

Data Type	Channels	Format	Trials	Rate (Hz)
No augmentation	22	22 × 875	576	250
Augmentation	22	22 × 875	6336	250

**Table 3 brainsci-12-01233-t003:** Comparison table of the proposed method with other methods.

	CSP-LDA (Baseline)	Ang et al. * [22]	Xie et al. [39]	Mahamune et al. [40]	Sakhavi et al. [41]	Qiao et al. [30]	Our Method
Dataset	2a (DA)	2a	2a	2a	2a	2a	2a (DA)
S1	78.2	76.0	81.8	87.1	87.5	89.1	88.3
S2	74.1	56.5	62.5	56.2	65.3	69.2	72.8
S3	88.5	81.3	88.8	93.0	90.3	89.5	93.7
S4	64.3	61.0	63.7	68.7	66.7	71.6	76.2
S5	50.2	55.0	62.9	39.8	62.5	64.1	60.2
S6	57.5	42.3	58.5	52.0	45.5	50.7	71.1
S7	85.8	82.8	86.6	89.9	89.8	89.2	84.1
S8	87.5	81.3	85.1	72.1	83.3	84.1	96.4
S9	71.7	70.8	90.0	82.6	79.5	82.1	83.7
AVG	73.1	67.8	75.5	71.2	74.5	76.6	80.7

## Data Availability

The EEG data used to validate the experimental results can be obtained from http://www.bbci.de/competition/iv/#download (accessed on 20 July 2022).

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
