# Peer review of "A Parallel Feature Fusion Network Combining GRU and CNN for Motor Imagery EEG Decoding"

_brainsci, 2022, doi:10.3390/brainsci12091233_

Round 1

Reviewer 1 Report

please mention the position of the 22 electrodes according to the standard which is in this link (http://www.brainm.com/software/pubs/dg/BA_10-20_ROI_Talairach/nearesteeg.htm) 

Why only 9 subjects are considered in this research? is it enough? why not more than nine?

Author Response

Many thanks for your valuable comments and suggestions on our manuscript entitled " A Parallel Feature Fusion Network Combining GRU and CNN for Motor Imagery EEG Decoding" (brainsci-1858010). Those comments are very helpful for revising and improving our paper. We have studied the comments carefully and made corrections which we hope meet with approval. The main corrections are in the manuscript and the responds to the reviewers’ comments are as follows (the replies are highlighted in red).

Point 1: Please mention the position of the 22 electrodes according to the standard which is in this link (http://www.brainm.com/software/pubs/dg/BA_10-20_ROI_Talairach/nearesteeg.htm)

Response 1: In line 333, we add the electrode montage corresponding to the international 10-20 system.

Point 2: Why only 9 subjects are considered in this research? is it enough? Why not more than nine?

Response 2: Currently, most of the studies based on deep learning methods to decode four classes of MI-EEG use the 2008 BCI competition dataset IV2a. This dataset records MI-EEG of 9 subjects. Section 3.1 of this paper describes this dataset in detail. In section 3.3 of this paper, we discuss several articles that use the same dataset. It is a very instructive observation that a large amount of EEG data can guarantee the performance of BCI. In future work, we will design experimental paradigms that use EEG caps to acquire 4-category MI signals from multiple volunteers for decoding tests.

Reviewer 2 Report

Major concerns

p2, line 117, "To solve the above problems, this study proposes a parallel GRU-CNN feature fusion network (GCFN) that takes the raw EEG data and time-frequency image as input data." First of all, this is an overstatement since many other publications with parallel architecture use both recursive neural networks and convolutional neural networks in the same architecture. 

More importantly, the authors did not explain why they decided to feed the time series to a recursive neural network and also the bandpass images of the wavelet coefficients through a 3D convolutional neural network (by the way, they should spell out CWT the first time it is used in the text). 

This is not a novel idea since the GRU-CNN model has been used previously (e.g., Weizheng Qiao and Xiaojun Bi. 2019. Deep Spatial-Temporal Neural Network for Classification of EEG-Based Motor Imagery. In Proceedings of the 2019 International Conference on Artificial Intelligence and Computer Science (AICS 2019). Association for Computing Machinery, New York, NY, USA, 265–272. https://doi.org/10.1145/3349341.3349414), and CWT-CNN was also used previously (e.g., Li, F.; He, F.; Wang, F.; Zhang, D.; Xia, Y.; Li, X. A Novel Simplified Convolutional Neural Network Classification Algorithm of Motor Imagery EEG Signals Based on Deep Learning. Appl. Sci. 2020, 10, 1605. https://doi.org/10.3390/app10051605. This paper claims 83.2% accuracy, which is higher than the manuscript under review). 

Why use both the raw time series and the wavelet transform? They have the same information content, except the wavelet presents the raw data in a fancier way. 

Finally, another complication is that the wavelet transform raises many more questions than answers. For example, which is the "optimal" mother wavelet (Haar, Daubechies, Biorthogonal, Coiflets, Symlets, Morlet, Mexican Hat, Meyer, etc.) for the data they used? How did the authors decide on Morlet? Is there anything intrinsically related to EEG that makes Morlet better than any other wavelets? 

Minor concerns

p1. GRU abbreviation should be explained in the abstract 

p2, line 112, "decoding methods do not fully utilize the features of EEG." What features are the authors referring to? 

p2, line 116, "decoding model fail to fully utilize the effective information of EEG data." What does it mean "the effective information of EEG"? How is that measured? How is that not "fully utilized" by existing methods?

p.3, line 135, why "a fifth-order Butterworth bandpass filter"?

p.3, line 141, what is CWT? 

p3., line 136, How was the filtered "standardized"?

p3, line 136, what does it mean " to eliminate the differences between features"?

p.4, line 162, "It can be seen from Figure 2 that raw MI-EEG data is usually composed of a 2D matrix in the form of C × T. The rows of the matrix store the data in the channels, and the columns store the data recorded at each sampling point." It seems that C = EEG channel number. On p.6, line 233,  "In the input part of EEG-GRU, this paper reorganizes the EEG data into the expression of T × C for input into the GRU network, where T corresponds to the time step of GRU, and C represents the feature number of each time step." Here C = feature number of each time step. Is this the same as the EEG channel number? If yes, why not say that? If not, maybe a different notation should be used. 

p.8, line 258, "In the input stage, the time-frequency image and EEG data are normalized" How were they normalized?

Reviewer 3 Report

1. The authors failed to show the novelty in their work. Please clarify the innovation in the study.
2. Please re-write the abstract as it looked a very simple one. Abstract should discuss three important things. (i). Limitations of available literature (ii) Method proposed by the author with technical information (iii) advantages and application of proposed method
3. I recommend authors to use multiscale principal component analysis (MSPCA) which is a combination of PCA and wavelets and useful for noise removal from network packets?
The details of MSPCA can be found in "Motor imagery BCI classification based on novel two-dimensional modelling in empirical wavelet transform"
4. For BCI system, signal decomposition methods always play significant role. I recommend authors to have a look on following article
"Motor Imagery EEG Signals Classification Based on Mode Amplitude and Frequency Components Using Empirical Wavelet Transform"
5. Did authors try to use non-linear features for correct identification in BCI? I recommend authors to include discussion of mean energy, mean Teager-Kaiser energy, SHANNON WAVELET ENTROPY and Log energy entropy.
6. The combination of signal decomposition with dimension reduction techniques along with neural networks can be one effective tool for both subject dependent and independent BCI frameworks. Authors need to discuss this issue; detail may be found in "Exploiting dimensionality reduction and neural network techniques for the development of expert brain-computer interfaces".
7. The authors recorded dataset from very few subjects. Is it possible to collect dataset from more subjects? If it is not possible, at least a discussion is needed for a framework tested on 58 subjects. See following article
"Towards the development of versatile brain-computer interfaces"
8. Please provide a comprehensive comparison of your study with the available literature in terms of classification accuracy, number of channels, features, and execution time with the following articles,
"A new framework for automatic detection of motor and mental imagery EEG signals for robust BCI systems", "A Matrix Determinant Feature Extraction Approach for Decoding Motor and Mental Imagery EEG in Subject Specific Tasks", "Motor imagery BCI classification based on novel two-dimensional modelling in empirical wavelet transform",
"Identification of Motor and Mental Imagery EEG in Two and Multiclass Subject-Dependent Tasks Using Successive Decomposition Index"
9. Please provide the details of future direction and possible solutions to continue this topic.
10. Finally, I suggest authors to sit with English native speaker to improve the writing of proposed work.

Round 2

Reviewer 3 Report

Authors revised article according to comments and I have no more questions